# Coronavirus Pseudotypes for All Circulating Human Coronaviruses for Quantification of Cross-Neutralizing Antibody Responses

**DOI:** 10.3390/v13081579

**Published:** 2021-08-10

**Authors:** Alexander Thomas Sampson, Jonathan Heeney, Diego Cantoni, Matteo Ferrari, Maria Suau Sans, Charlotte George, Cecilia Di Genova, Martin Mayora Neto, Sebastian Einhauser, Benedikt Asbach, Ralf Wagner, Helen Baxendale, Nigel Temperton, George Carnell

**Affiliations:** 1Laboratory of Viral Zoonotics, Department of Veterinary Medicine, University of Cambridge, Cambridge CB3 0ES, UK; jlh66@cam.ac.uk (J.H.); matteo@diosvax.com (M.F.); ms2616@cam.ac.uk (M.S.S.); clg64@cam.ac.uk (C.G.); gwc26@cam.ac.uk (G.C.); 2DIOSynVax Ltd., Cambridge CB3 0ES, UK; 3Viral Pseudotype Unit, University of Kent, Chatham ME4 4TB, UK; D.Cantoni@kent.ac.uk (D.C.); cd516@kent.ac.uk (C.D.G.); M.Mayora-Neto@kent.ac.uk (M.M.N.); N.Temperton@kent.ac.uk (N.T.); 4Institute for Medical Microbiology and Hygiene, University of Regensburg, 93053 Regensburg, Germany; Sebastian.Einhauser@klinik.uni-regensburg.de (S.E.); benedikt.asbach@klinik.uni-regensburg.de (B.A.); Ralf.Wagner@klinik.uni-regensburg.de (R.W.); 5Institute for Clinical Microbiology and Hygiene, University Hospital, 93053 Regensburg, Germany; 6Royal Papworth Hospital NHS Foundation Trust, Cambridge CB2 0AY, UK; hbaxendale@nhs.net

**Keywords:** SARS-CoV-2, COVID-19, coronavirus, pseudotyped virus, neutralization

## Abstract

The novel coronavirus SARS-CoV-2 is the seventh identified human coronavirus. Understanding the extent of pre-existing immunity induced by seropositivity to endemic seasonal coronaviruses and the impact of cross-reactivity on COVID-19 disease progression remains a key research question in immunity to SARS-CoV-2 and the immunopathology of COVID-2019 disease. This paper describes a panel of lentiviral pseudotypes bearing the spike (S) proteins for each of the seven human coronaviruses (HCoVs), generated under similar conditions optimized for high titre production allowing a high-throughput investigation of antibody neutralization breadth. Optimal production conditions and most readily available permissive target cell lines were determined for spike-mediated entry by each HCoV pseudotype: SARS-CoV-1, SARS-CoV-2 and HCoV-NL63 best transduced HEK293T/17 cells transfected with ACE2 and TMPRSS2, HCoV-229E and MERS-CoV preferentially entered HUH7 cells, and CHO cells were most permissive for the seasonal betacoronavirus HCoV-HKU1. Entry of ACE2 using pseudotypes was enhanced by ACE2 and TMPRSS2 expression in target cells, whilst TMPRSS2 transfection rendered HEK293T/17 cells permissive for HCoV-HKU1 and HCoV-OC43 entry. Additionally, pseudotype viruses were produced bearing additional coronavirus surface proteins, including the SARS-CoV-2 Envelope (E) and Membrane (M) proteins and HCoV-OC43/HCoV-HKU1 Haemagglutinin-Esterase (HE) proteins. This panel of lentiviral pseudotypes provides a safe, rapidly quantifiable and high-throughput tool for serological comparison of pan-coronavirus neutralizing responses; this can be used to elucidate antibody dynamics against individual coronaviruses and the effects of antibody cross-reactivity on clinical outcome following natural infection or vaccination.

## 1. Introduction

Since its emergence in 2019 [1], the novel coronavirus SARS-CoV-2 (Severe Acute Respiratory Syndrome Coronavirus 2) has been responsible for over 190 million cases of Coronavirus Disease 2019 (COVID-19) worldwide. Infection triggered immunopathology causes fatal sequalae, most often in individuals in the later decades of life, with deaths now exceeding four million [2]. SARS-CoV-2 is the seventh identified human coronavirus (HCoV) [3]; the early metagenomic sequencing of virus isolates identified SARS-CoV-2 as a member of the betacoronavirus genus of *Coronavirinae* in the *Coronaviridae* family, closely related to other high pathogenicity human coronaviruses such as SARS-CoV-1 (Severe Acute Respiratory Syndrome Coronavirus 1) and MERS-CoV (Middle East Respiratory Syndrome Coronavirus) [1,4] (Figure 1 and Figure 2). In addition, four seasonal human coronaviruses (HCoV-229E, HCoV-NL63, HCoV-HKU1 and HCoV-OC43, also collectively known as the Common Cold Coronaviruses (CCCoVs) or Endemic Human Coronaviruses) associated with low morbidity respiratory infections in immunocompetent children and adults have been identified [5]. The CCCoVs are endemic, frequently first encountered in the first decade of life. They are isolated seasonally from a substantial proportion of respiratory diseases in adults, suggesting a highly prevalent re-exposure and short-lived immune protection despite a widespread seropositivity [5,6,7,8]. Throughout the SARS-CoV-2 pandemic, there have been questions regarding the extent of pre-existing immunity to SARS-CoV-2 induced by prior exposure to seasonal coronaviruses and the impact of cross-reactivity on disease progression [9]. Furthermore, understanding the dynamics of antibody responses observed after the SARS-CoV-1 pandemic, MERS-CoV outbreaks or a following seasonal coronavirus infection may help define the durability of protection induced by vaccination or natural infection with SARS-CoV-2 [8]. 

The *Coronaviridae* are a family of enveloped positive sense ssRNA viruses with an approximately 27–32 kB genome size, named after the morphology of their characteristic spike (S) envelope glycoproteins trimers used for entry into host cells [10]. The S proteins of all human coronaviruses consist of an S1 subunit containing a receptor-binding domain (RBD) and an S2 subunit that mediates membrane fusion after receptor binding [11,12]; despite this structural similarity, the S proteins of the seven human coronaviruses utilise a diverse set of host cell receptors for entry [13,14,15] (Table 1). 

The S protein contains several important neutralizing epitopes characterised for SARS-CoV-1 [16], SARS-CoV-2 [17,18,19,20] and MERS-CoV [21,22], and is the major focus of most vaccine development strategies for SARS-CoV-2 [23,24]. In addition, many HCoVs feature a requirement for additional host entry factors, including cellular proteases such as TMPRSS2, which mediates proteolytic priming of the SARS-CoV-2 S protein [25,26]. The two endemic betacoronaviruses HCoV-OC43 and HCoV-HKU1 additionally harbour surface Haemagglutinin-Esterase (HE) proteins, which are thought to act as receptor-destroying enzymes by cleaving sialic acid as in Influenza C and D [27,28]. Other surface proteins such as the Envelope (E) and Membrane (M) proteins are shared by all human coronaviruses [29] and have roles in virion assembly and maturation [29,30,31]. Various studies using ELISA, Lateral Flow assays and flow cytometry have identified a cross-reactive antibody binding to the S proteins of multiple coronaviruses [32,33,34,35]. However, since antibody binding does not always correlate with protection from clinical disease and due to the theoretical risk of disease enhancement by non-neutralizing antibodies [36,37], it is essential to investigate whether these antibody responses are capable of cross-protection by neutralizing virus entry into host cells [35]. Elucidating the determinants of cross-neutralization has so far been impeded by the difficulties of the in vitro culture of wild-type coronaviruses, limited investigation of neutralizing antibody responses against seasonal coronaviruses and the intrinsic challenges of safely handling the high pathogenicity coronaviruses [32]. 

Since its emergence, 18 vaccines against SARS-CoV-2 have been approved and over 3,600,000,000 doses administered globally, with a further 104 vaccine candidates undergoing clinical trials and 184 in preclinical development stages [2,38]. A number of SARS-CoV-2 variants have now emerged; of these, four variants have been classified by the World Health Organisation (WHO) as Variants of Concern (VOCs) due to an association with increased transmissibility, virulence or immune escape [39,40,41]—these variants have been labelled Alpha/B.1.1.7, Beta/B.1.351, Gamma/P.1, and Delta/B.1.617.2 by the WHO or Pango lineage naming systems, respectively. The ability of vaccines or prior exposure to protect against these VOCs and the durability of vaccine responses is an urgent focus of ongoing research.

Pseudotype viruses (PVs) offer a safe and effective surrogate to handling wild-type viruses [42,43]. These typically consist of a replication-defective viral core (usually based on the HIV, MLV or VSV capsid), which packages a reporter gene (most often Firefly Luciferase or GFP reporter genes) during assembly and acquires a lipid envelope from producer cells studded with heterologous viral glycoproteins during budding [42,44]. These PVs mimic authentic glycoprotein-mediated entry into cells, leading to a measurable output from the reporter gene [42,43]. The production of PVs ‘pseudotyped’ with the coronavirus S proteins can, therefore, be used to identify essential host entry factors necessary for coronavirus entry, to assay the neutralization of human coronaviruses by antibodies directed against S protein epitopes or to screen for an antiviral inhibition of entry [25,45,46]. These assays can be performed safely (at BSL2) outside high containment, avoiding virus culture acquired in vitro mutations during cell culture passage of wild-type viruses. Pseudotyped viruses have been used extensively to study the neutralization of high pathogenicity viruses such as Ebola, Marburg, Lassa Fever and Rabies [43,47,48], including investigating strain-specific and cross-reactive neutralizing antibodies against Influenza A [42,49]. A number of pseudotype systems have been utilised in the past to study human coronaviruses—in particular, PVs have been used extensively to study the neutralization of SARS-CoV-2 [45,46,50,51]; however, there is a need for a unified panel of PVs for each human coronavirus produced under similar conditions at high production titres sufficient to allow high-throughput comparisons of cross-reactive neutralizing immune responses.

Here, we present a complete panel of lentiviral particles pseudotyped with the S proteins for all seven human coronaviruses as a tool to assay the breadth of neutralizing responses against SARS-CoV-2 and other coronaviruses. Production was optimised to achieve high titres using the same lentiviral luciferase system for each human coronavirus pseudotype, allowing comparable investigations in parallel across the panel. These pseudotypes can be used to investigate the breadth of the neutralization induced by natural infection or vaccination, to further investigate antibody dynamics against individual human coronaviruses such as the durability of protection and to assess the impact of prior immunity on disease progression and vaccine response.

## 2. Materials and Methods

### 2.1. Plasmids

The codon-optimised S genes for all seven human coronaviruses, the SARS-CoV-2 E and M genes and the HCoV-OC43 and HCoV-HKU1 HE genes were synthesised commercially by GeneArt and cloned into the pEVAC vector plasmid [52] at the University of Regensburg. Additionally, the four HCoV seasonal S genes were synthesised in a pcDNA3.1+ backbone in parallel by GeneArt Gene Synthesis, Thermo Fisher, purchased by Xiao-Ning Xu at Imperial College London. The human ACE2 receptor plasmid pCAGGS-ACE2 and the human TMPRSS2 protease encoding pCAGGS-TMPRSS2 plasmid were provided by S Pöhlmann and M Hoffman at the Leibniz Institute for Primate Research. The lentiviral packaging plasmid p8.91 [53] and firefly luciferase reporter plasmid pCSFLW [54] were used for pseudotype production [42].

### 2.2. Cell Lines

HEK293T/17 and CHO-K1 cells were obtained from E. Wright at the University of Sussex; MDCK-II cells were obtained from M. Schwemmle at the University of Freiburg, and VeroE6 cells were obtained from Professor Juan Carlos de la Torre at Scripps Research. Cells were cultured at 37 °C with 5% CO_2_ using Dulbecco’s Modified Eagle Medium (DMEM) GlutaMAX™ (Gibco™) supplemented with 10% Foetal Calf Serum (Invitrogen) and 1% Penicillin–Streptomycin (Gibco™), referred to as Complete DMEM.

### 2.3. Production of Lentiviral Pseudotypes for Human Coronaviruses

Pseudotyped viruses were produced in HEK293T/17 cells. HEK293T/17 producer cells were sub-cultured in 6-well Delta Nunc plates (Thermo Fisher Scientific, Waltham, MA, USA) and co-transfected at 60% confluence with 250 ng of the p8.91 lentiviral packaging plasmid, 375 ng of the pCSFLW firefly luciferase reporter plasmid and one or more plasmids encoding viral surface proteins per well. The transfection complex was prepared in 100 µL Opti-MEM™ (Thermo Fisher Scientific, Loughborough, UK) using 0.3 µL FuGENE^^®^^-HD (Promega, Madison, WI, USA) transfection reagent per 100 ng of plasmid. The cell supernatant was collected after 48 or 72 h using sterile 2.5 mL syringes and passed through a 0.45 µm cellulose acetate or polyethersulfone (PES) filter (Millipore Sigma, Burlington, MA, USA) prior to storage in microcentrifuge tubes at −80 °C. HCoV-HKU1 lentiviral PVs were produced in T75 flasks (Thermo Fisher Scientific, Waltham, MA, USA) with 1 µg of HCoV-HKU1 spike pcDNA3.1, 1.5 µg pCSFLW and 1 µg p8.91 lentiviral packaging plasmid. The following day, 1.5 U of exogenous neuraminidase were added to the cell supernatant. The cell supernatant was collected after 48 h, as described. 

### 2.4. Titration of Lentiviral Pseudotypes

Titration of PVs was performed by transduction of target cells and measurement of luminescence or fluorescence from target cells following PV entry. Target cells were sub-cultured in T25 flasks (Thermo Fisher Scientific, Waltham, MA, USA); transfection with plasmids encoding entry factors was performed 24 h prior to titration using 300 µL Opti-MEM™ and 0.3 µL FuGENE^®^-HD transfection reagent per 100 ng of plasmid. Serial dilutions of 100 µL of pseudotype-containing cell supernatant were prepared in complete DMEM in white flat-bottom 96-well Nunclon© plates (Thermo Fisher Scientific, Waltham, MA, USA). Target cells were detached and approximately 1.5 × 10^4^ cells in 50 µL complete DMEM were added per well, prior to incubation in a humidified cell-culture incubator for 48 h at 37 °C in 5% CO_2_. Equal volumes of Bright-Glo™ reagent and Phosphate Buffered Saline (PBS) were mixed and 25 µL added to each well; luminescence output was measured using a GloMax™ 96-well plate luminometer (Promega, Madison, WI, USA) after 5 min incubation at room temperature.

### 2.5. Pseudotype Based Microneutralization Assays

Pseudotype based microneutralization (pMN) assays were performed as described by us previously [42,55,56,57]. Human sera from SARS-CoV-2 seropositive and seronegative health workers and patients hospitalised with COVID-19 was supplied by Papworth General Hospital through the Humoral Immune Correlates for COVID-19 consortium (https://www.hicc-consortium.com/, accessed on 10 May 2021) [58]; seropositivity was determined by ELISA and Luminex binding assays. Reference sera 20/130 and 20/162 were obtained from the National Institute for Biological Standards and Control (NIBSC). A dilution series of human sera was prepared in a white flat bottom 96-well Nunclon© plate, and 50 µL of PV containing supernatant diluted in complete DMEM to give 1 × 10^6^ RLU equivalent was added per well. The plate was then centrifuged at 300× *g* for 3 min prior to incubation in a humidified cell-culture incubator for 1 h at 37 °C, 5% CO_2_. Sub-cultured target cells were added prior to incubation for 48 h and luminescence measurement as described above. Analysis was performed by non-linear regression after normalisation to 100% and 0% neutralization utilizing plate controls.

### 2.6. Statistical Analysis

Statistical analysis was performed where possible to determine whether differences in HCoV PV titre were significant. D’Agostino–Pearson normality tests, one way ANOVA tests with Tukey’s Multiple Comparison tests, Kruskal–Wallis tests with Dunn’s multiple comparison tests, Mann–Whitney tests and unpaired t-tests were performed (see Appendix A for details). *p*-values were set at *p* ≤ 0.05 (*), <0.005 (**), <0.0005 (***) and <0.00005 (****). 

## 3. Results

### 3.1. Optimising Production of Coronavirus Spike-Bearing Pseudotype Viruses

High production titres of lentiviral pseudotypes were generated for all seven human coronavirus spike proteins via optimisation of production protocol and transfection of permissive cell lines with necessary entry factors. First, plasmids encoding the codon-optimised full length spike gene were generated for each of the human coronaviruses and the optimal input of the spike plasmid for transfection into producer HEK293T/17 cells was determined (Figure 3a). Spike plasmids were co-transfected into producer cells alongside the lentiviral gag-pol p8.91 expression plasmid and the firefly luciferase reporter pCSFLW expression plasmid using FuGENE-HD transfection reagent (Figure 4). Collection of cell supernatant at 72 h after transfection was found to harvest approximately 9× higher titres of virus than collection at 48 h; pre-transfection of producer cells with the S gene 24 h before transfection with the lentiviral luciferase plasmid system did not boost production (Figure 3b).

HCoV-HKU1 PVs were generated successfully by the addition of exogenous neuraminidase 24 h after transfection, resulting in production titres of approximately 1 × 10^7^ RLU/mL when transducing CHO-K1 target cells (Figure 5a). Functional titres were not observed when PVs were produced without the addition of exogenous neuraminidase (Figure 5a). Production titres of HCoV-OC43 were enhanced by the co-transfection of TMPRSS2 into the producer cell (Figure 5c), whilst the transfection of the haemagglutinin-esterase (HE) protein did not increase titres (Figure 5b). We also generated SARS-CoV-2 pseudotypes bearing the S protein in addition to either the E protein, the M protein or with different ratios of both the E and M proteins (Figure 6).

### 3.2. Permissible Cell Lines Were Identified and Transfected with Cellular Factors to Enhance Entry

To identify the most permissive cell lines for spike-mediated entry by each human coronavirus pseudotype, the panel of lentiviral pseudotypes was used to infect a range of different cell culture lines—each PV was used to infect HEK293T/17, HUH7, VeroE6, MDCK-II and A549 cells. The PVs pseudotyped with the S proteins from ACE2 using viruses (SARS-CoV-1, SARS-CoV-2 and HCoV-NL63) entered HEK293T/17 and HUH7 cells efficiently with similarly high titres, whilst HCoV-MERS and HCoV-229E preferentially entered HUH7 cells (Figure 7a,b). There was no entry by any HCoV PV into A549, MDCK-II and VeroE6 cells. The sialic-acid-using betacoronavirus HCoV-HKU1 was subsequently found to enter CHO-K1 cells (Figure 5a). 

We then sought to investigate whether entry into target cells could be increased by transient transfection with entry factors previously identified to be necessary for entry by wild-type human coronaviruses. The ACE2 receptor was identified as the main cellular receptor necessary for entry of the sarbecoviruses, including SARS-CoV-1 and SARS-CoV-2 and alphacoronaviruses such as HCoV-NL63 [26,29]. The transfection of ACE2 into HEK293T/17 cells boosted entry by over a 100-fold for HCoV-NL63, by a factor of 30 for SARS-CoV-2 and 5-fold for SARS-CoV-1. The cellular protease TMPRSS2 was also identified as an important entry factor permitting the endosomal entry of a number of human coronaviruses [25]; whilst the transfection of TMPRSS2 alone had little effect on titres, co-transfection of TMPRSS2 as well as ACE2 led to an approximately 300-fold increase in entry titre for HCoV-NL63, 3000-fold for SARS-CoV-2 and by a factor of 40 for SARS-CoV-1. The optimal ratio of these entry factors was determined for each of SARS-CoV-1, SARS-CoV-2 and HCoV-NL63 (Figure 8a). In addition, the transfection of TMPRSS2 in HEK293T/17 cells rendered this cell line permissible for high-titre entry by HCoV-HKU1 and enhanced entry by HCoV-OC43, representing an alternative cell line in addition to the use of CHO-K1 cells (Figure 8b).

### 3.3. The Panel of PVs Can Be Used to Assess HCoV Neutralization in Human Serum Samples

We subsequently sought to investigate whether the panel of luciferase-bearing lentiviral pseudotyped viruses could be used to assess the neutralization of HCoV PVs by human sera. PVs were incubated for an hour at 37 °C with human sera prior to the addition of target cells; the neutralization of entry was assayed as a reduction in luciferase output after 48 h. Serum samples from five SARS-CoV-2 seronegative individuals, five SARS-CoV-2 seropositive individuals, five COVID-19 patients and two SARS-CoV-2 seropositive NIBSC reference sera were used to neutralize SARS-CoV-2, HCoV-HKU1, HCoV-NL63 and HCoV-229E PVs. The neutralization curves of individual samples are shown in Figure 9a (further neutralization curves from this dataset are available in Appendix A), allowing the identification of samples able to cross-neutralize multiple human coronaviruses. The IC50 values for each sub-group against these PVs are shown in Figure 9b, allowing the comparison of neutralization between groups.

By measuring the neutralization ability of sera from eight COVID-19 patients at sequential time-points, we were also able to investigate whether the neutralization of other human coronaviruses such as HCoV-NL63 changed during the course of infection with SARS-CoV-2 (see Figure 10). 

## 4. Discussion

This paper presents a panel of lentiviral PVs pseudotyped with S proteins from each of the seven human coronaviruses. In Figure 9, we demonstrated that our panel can be used to investigate the neutralization of different HCoVs by human serum samples—this could be used to characterise an individual’s ability to neutralize multiple human coronaviruses (Figure 9a), to compare the neutralization ability between sub-groups in a cohort of serum samples (Figure 9b) or to study changes in the neutralization ability over time (Figure 10). This allows a further investigation of responses to individual HCoVs, including the assessment of antibody dynamics and the longevity of protection, and can be used to investigate cross-neutralizing responses across the human coronaviruses.

The production of all HCoV PVs using the same system under similar production conditions means the investigation of neutralizing responses can occur in parallel providing directly comparable results. We also achieved high titres of at least 1 × 10^7−8^ RLU/mL for all HCoV PVs; since an input of 1 × 10^5−6^ RLU/well of a 96-well plate was recommended for optimal pMN assay, our titres were sufficient for comparisons of large serum cohorts using the same preparation of virus. In contrast to the neutralization assays with wild-type viruses, the use of PVs reduces the risk of inadvertently selecting for cell-culture-associated spike variants during repeated passage in tissue culture; on the other hand, PVs bearing circulating variants such as the known Variants of Concern can easily be generated via site directed mutagenesis [59]. Our panel, therefore, represents a useful tool for studying neutralizing responses in cohorts of naturally infected or vaccinated individuals, whilst avoiding the difficulties associated with in vitro culturing of wild-type coronaviruses or the challenges of safely handling high pathogenicity coronaviruses. Limitations of this assay include the difficulty standardizing the transfection of entry factors into target cells between experiments and the comparative high labour input of neutralization assays using transduction of cells in tissue culture relative to more easily automatable antibody binding assays.

The ability to compare the neutralization of all HCoVs will help answer critical research questions during the next phases of the SARS-CoV-2 pandemic. Firstly, whilst there is clear evidence of cross-reactivity of antibody binding to multiple coronaviruses—for example, leading to apparent SARS-CoV-2 seropositivity in pre-pandemic samples—it is still undetermined whether these responses were likely to be protective, or indeed which of the identified human coronaviruses are most likely responsible. Whilst evidence supporting an antibody dependent enhancement effect is weak, there is a theoretical risk of enhanced disease severity following the recruitment of non-neutralizing binding antibodies induced by prior exposure to seasonal coronaviruses; elucidating whether cross-reactive responses are capable of neutralization is, therefore, a critical research priority. Our panel can also aid in the investigation into the antibody neutralization of individual HCoVs; identifying the longevity of high neutralizing antibody titres against the seasonal HCoVs or in serum samples from the SARS-CoV-1 pandemic and MERS-CoV outbreaks could inform our estimates for the persistence of SARS-CoV-2 protection following infection or the durability of neutralizing responses following vaccination. The strong correlation between the SARS-CoV-2 neutralization titre and protection from clinical disease means that our panel of PVs could be a useful tool for identifying correlates of immunity against SARS-CoV-2, especially in comparison to serological assays for antibody binding such as ELISA [60]. Furthermore, our panel of PVs could be used to identify additional cellular factors necessary for coronavirus entry, or for screening the breadth of antiviral inhibitors of coronavirus entry.

The HCoV PVs differed in their cellular tropism; the permissibility of different cell lines for S-protein pseudotyped PV transduction likely depended on the different expression levels of cellular receptors as well as the intrinsic capacity for supporting lentiviral infection and reporter expression. For example, HUH7 cells express DPP4 and Aminopeptidase N at higher levels than HEK293T/17 cells, presumably explaining the higher entry titres by MERS-CoV and HCoV-229E. The transfection of ACE2 and TMPRSS2 boosted entry of the ACE2 using PVs—noticeably, this effect was greater in HEK293T/17 cells than HUH7 cells. Since entry was similar in both un-transfected cell lines for HCoV-NL63 and SARS-CoV-2, it is possible that the preferential entry of HEK293T/17 cells was due to a higher ACE2/TMPRSS2 plasmid transfection efficiency in this cell line. Notably, entry of PVs into cell lines did not always reflect tropism of wild-type viruses; this is most evident in the minimal entry of all PVs into VeroE6 cells, which are most commonly used for culturing SARS-CoV-1 and SARS-CoV-2. This is likely explained by a higher expression of lentiviral restriction factors such as TRIM5α in this cell line compared to HEK293T/17 or HUH7 cells [61]. The efficiency of transduction of different cell lines, therefore, relates both to the S-protein tropism and expression of entry factors necessary for glycoprotein mediated entry, as well as the ability to support vector entry and reporter expression. 

A number of PVs featured enhanced entry following TMPRSS2 transfection into target cells. The role of TMPRSS2 in wild-type virus entry was primarily thought to be in facilitating endosomal cleavage of the S protein; additional roles have been suggested in the inactivation of antiviral restriction factors such as IFITM3, known to inhibit S-mediated entry [62]. The transfection boosted entry of SARS-CoV-1, SARS-CoV-2, HCoV-NL63, HCoV-OC43 and HCoV-HKU1; no increase was observed for HCoV-229E or HCoV-MERS using up to 75 ng TMPRSS2—however, since this protease has been reported to enhance entry of the wild-type viruses, it is possible that higher inputs could increase entry titres [63,64]. Titres of HCoV-OC43 were increased following the target cell TMPRSS2 transfection but were highest following the transfection of TMPRSS2 during production; this may indicate a requirement for S protein maturation prior to endocytosis. A number of other putative receptors for SARS-CoV-2 have since been identified in addition to ACE2—for example, neuropilin-1 [65]; it is possible that the transient expression of these receptors could likewise boost entry of SARS-CoV-2 PVs.

In conclusion, this panel provided a valuable tool to investigate neutralizing responses to SARS-CoV-2 and the other human coronaviruses, allowing key research questions regarding SARS-CoV-2 protection, human coronavirus cross-neutralization and COVID-19 immunopathology to be addressed.

## Figures and Tables

**Figure 1 viruses-13-01579-f001:**
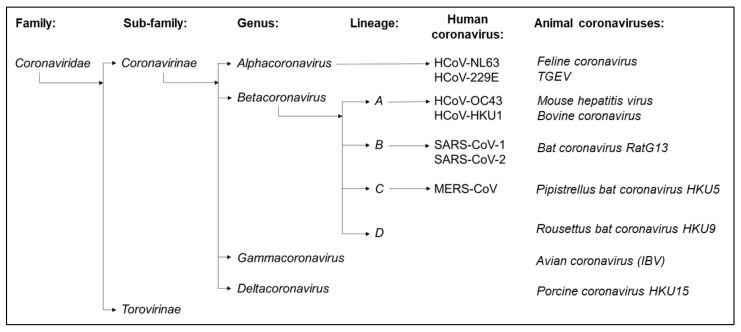
The seven human coronaviruses belong to either the alphacoronavirus or betacoronavirus genus of the Coronavirinae sub-family of the Coronaviridae. The alphacoronaviruses include seasonal coronaviruses HCoV-229E and HCoV-NL63, whilst the betacoronaviruses include high pathogenicity coronaviruses such as SARS-CoV-1, SARS-CoV-2 and MERS-CoV in addition to seasonal coronaviruses HCoV-HKU1 and HCoV-OC43. The Coronavirinae are ubiquitous pathogens of mammalian and avian species; each of the seven human coronaviruses emerged from independent zoonotic spillover events.

**Figure 2 viruses-13-01579-f002:**
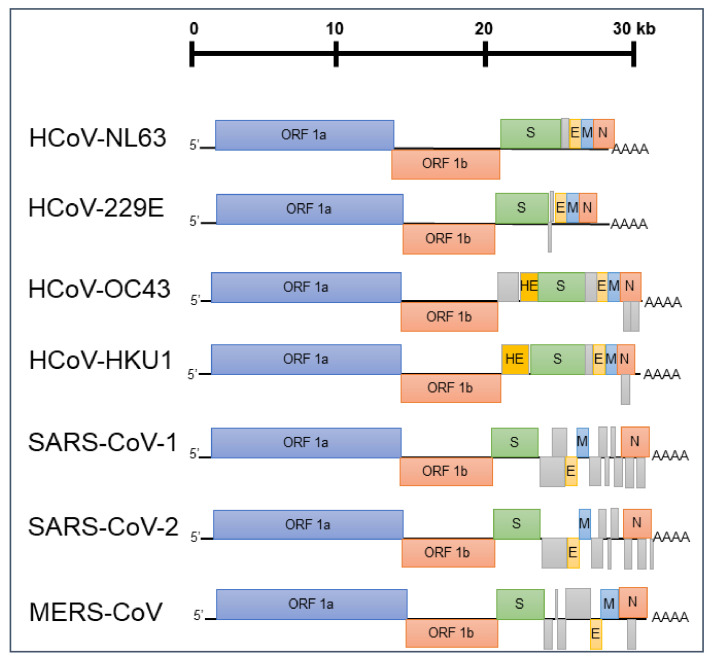
The genomic organization of the seven human coronaviruses: all coronaviruses encode two major Open Reading Frames (ORFs) known as ORF1a and ORF1b in addition to Spike (S), Envelope (E), Membrane (M) and Nucleocapsid (N) proteins. S proteins form trimeric envelope glycoproteins utilized for entry after receptor binding, whilst other surface proteins (E and M) have roles in viral assembly. All HCoVs also encode a variable set of non-structural proteins (shown in grey), and the two seasonal betacoronaviruses HCoV-HKU1 and HCoV-OC43 additionally encode surface Haemagglutinin-Esterase (HE) proteins as receptor-destroying enzymes.

**Figure 3 viruses-13-01579-f003:**
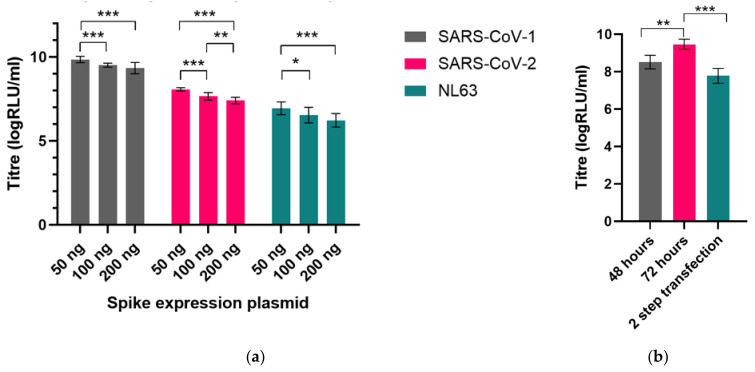
Optimising production of pseudotyped viruses: (**a**) Various amounts of spike expression plasmid were transfected for each human coronavirus, shown here with SARS-CoV-1, SARS-CoV-2 and HCoV-NL63. High titres were achieved for the entire panel using 50 ng of the pEVAC expression plasmid alongside 250 ng of p8.91 and 375 ng of pCSFLW. (**b**) SARS-CoV-2 PVs were harvested at either 48 h or 72 h after transfection; collection of PV containing supernatant at 72 h led to increased titres. 2-step transfection, with initial transfection of the S gene followed 24 h later by the lentiviral gag-pol and luciferase reporter plasmids and collection 72 h after the first transfection, led to a reduction in titre. Significant differences in titre shown by asterisks: * represents *p* ≤ 0.05, ** *p* ≤ 0.005 and *** *p* ≤ 0.0005, respectively.

**Figure 4 viruses-13-01579-f004:**
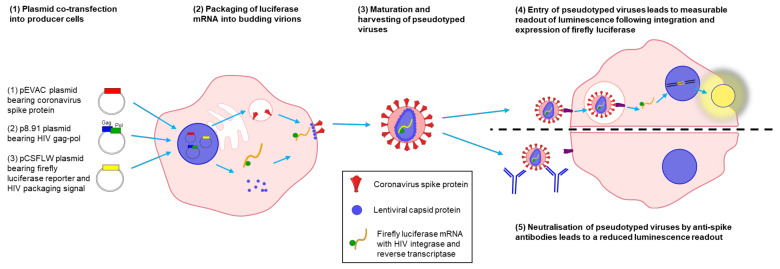
Schematic representing pseudotyped virus (PV) production and use in PV neutralization assays. Producer cells were co-transfected with plasmids encoding a coronavirus spike protein, a lentiviral gag-pol construct and the pCSFLW firefly luciferase reporter. The lentiviral capsid assembled into a PV core and packaged the pCSFLW reporter construct (consisting of the firefly luciferase coding sequence flanked by lentiviral LTRs), before budding from the plasma membrane and acquiring an outer membrane studded with coronavirus spike proteins. After spike-mediated entry into target cells, the pCSFLW reporter was integrated and expressed, leading to measurable luminescence from successfully transduced target cells. Neutralization of PVs by anti-spike antibodies following incubation with human sera can be measured as a reduction in luminescence output after target cell transduction.

**Figure 5 viruses-13-01579-f005:**
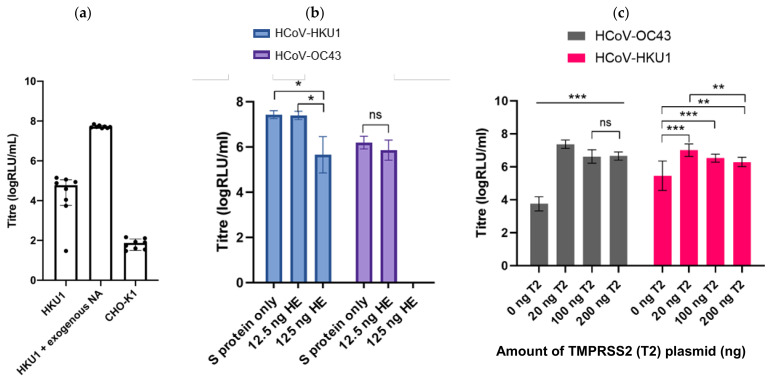
Enhancing titres of HCoV-OC43 and HCoV-HKU1: (**a**) entry of HKU1 PVs into CHO cells was enhanced after incubation with exogenous NA relative to CHO-K1 cell only control; (**b**) HCoV-OC43 or HCoV-HKU1 PVs were generated incorporating their corresponding Haemagglutinin Esterase (HE) protein in addition to the spike protein. Co-transfection of 12.5 ng of HE led to comparable titres to spike-only PVs for both viruses when used with TMPRSS2-transfected HEK293T/17 cells (see below). Transfection of higher inputs of HE led to reduced titres for both viruses. (**c**) Production of HCoV-OC43 and HCoV-HKU1 was enhanced by co-transfection of producer cells with TMPRSS2; this enhanced entry of HCoV-OC43 over 3000-fold compared to un-transfected HEK293T/17 cells. Significant differences in titre shown by asterisks: * represents *p* ≤ 0.05, ** *p* ≤ 0.005 and *** *p* ≤ 0.0005, respectively.

**Figure 6 viruses-13-01579-f006:**
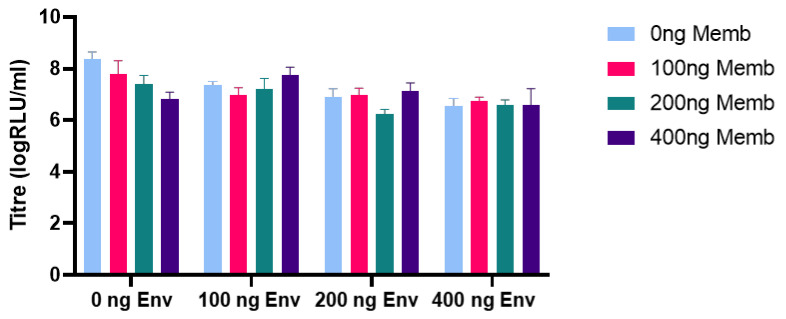
PVs were pseudotyped with the SARS-CoV-2 S protein in addition to combinations of the Envelope (Env) and Membrane (Memb) proteins; PVs produced with 100 ng Envelope and 400 ng Membrane in addition to 50 ng spike had the second highest titres relative to spike only.

**Figure 7 viruses-13-01579-f007:**
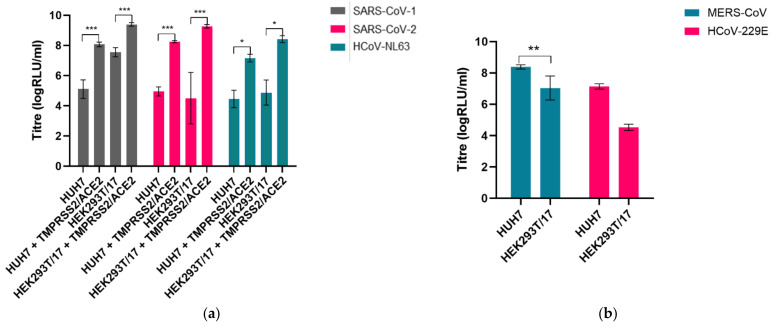
Entry of PVs was assessed by titration into various cell lines. (**a**) Entry of SARS-CoV-1 was highest in HEK293T/17 cells, whilst entry of SARS-CoV-2 and HCoV-NL63 was comparable in both HEK293T/17 and HUH7 cells. (**b**) MERS-CoV and HCoV-229E preferentially entered HUH7 cells compared to HEK293T/17 cells. Significant differences in titre shown by asterisks: * represents *p* ≤ 0.05, ** *p* ≤ 0.005 and *** *p* ≤ 0.0005, respectively.

**Figure 8 viruses-13-01579-f008:**
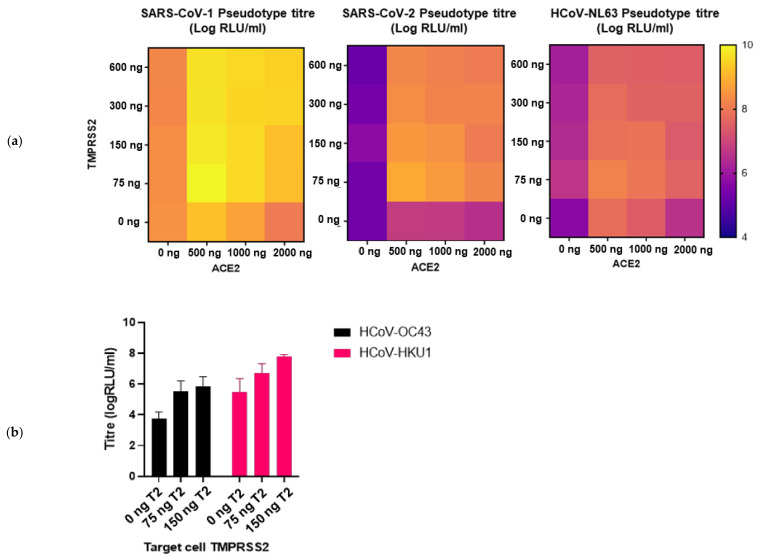
Enhancement of PV entry by target cell transfection: (**a**) Transfection of either HEK293T/17 cells or HUH7 cells with TMPRSS2 and ACE2 considerably boosted entry of the ACE2-using SARS-CoV-1, SARS-CoV-2 and HCoV-NL63 PVs. The optimal combination of ACE2 and TMPRSS2 for transfection into HEK293T/17 cells was determined for each PV—75 ng of TMPRSS2 expression plasmid and 500 ng of ACE2 expression plasmid supported the highest entry. (**b**) Entry of HCoV-HKU1 and HCoV-OC43 was enhanced by transfection of HEK293T/17 cells with TMPRSS2.

**Figure 9 viruses-13-01579-f009:**
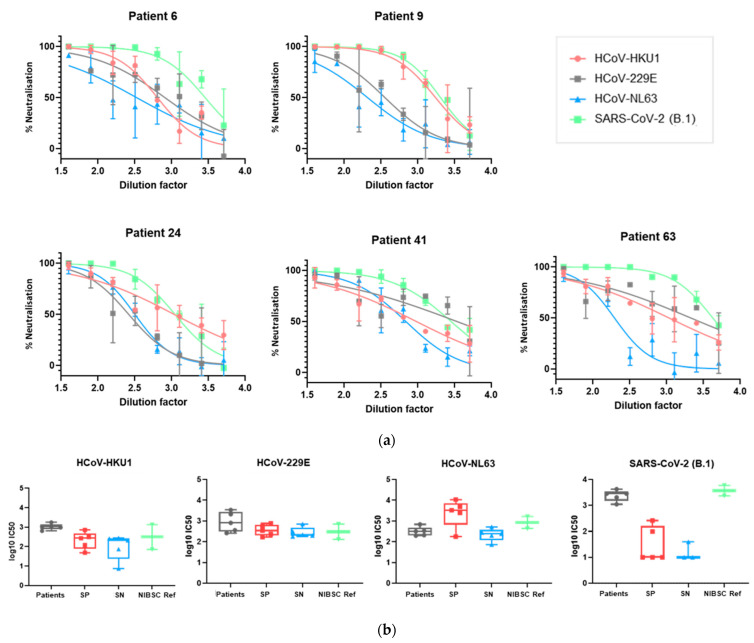
Neutralization of SARS-CoV-2, HCoV-NL63, HCoV-229E and HCoV-HKU1 by serum samples from 5 SARS-CoV-2 seropositive individuals (SP), 5 SARS-CoV-2 seronegative individuals (SN), 5 COVID-19 patients (patients) and 2 SARS-CoV-2 seropositive NIBSC reference sera (NISBC ref); (**a**) neutralization of SARS-CoV-2, HCoV-HKU1, HCoV-229E and HCoV-NL63 by 5 COVID-19 patient serum samples. Further neutralization curves by additional serum samples are shown in Appendix A. Curves fit by non-linear regression using GraphPad Prism 9; (**b**) IC50 values compared between each group of serum samples for each PV. IC50 values were calculated using GraphPad Prism 9 as the dilution of sera needed for 50% neutralization of luciferase output/PV entry.

**Figure 10 viruses-13-01579-f010:**
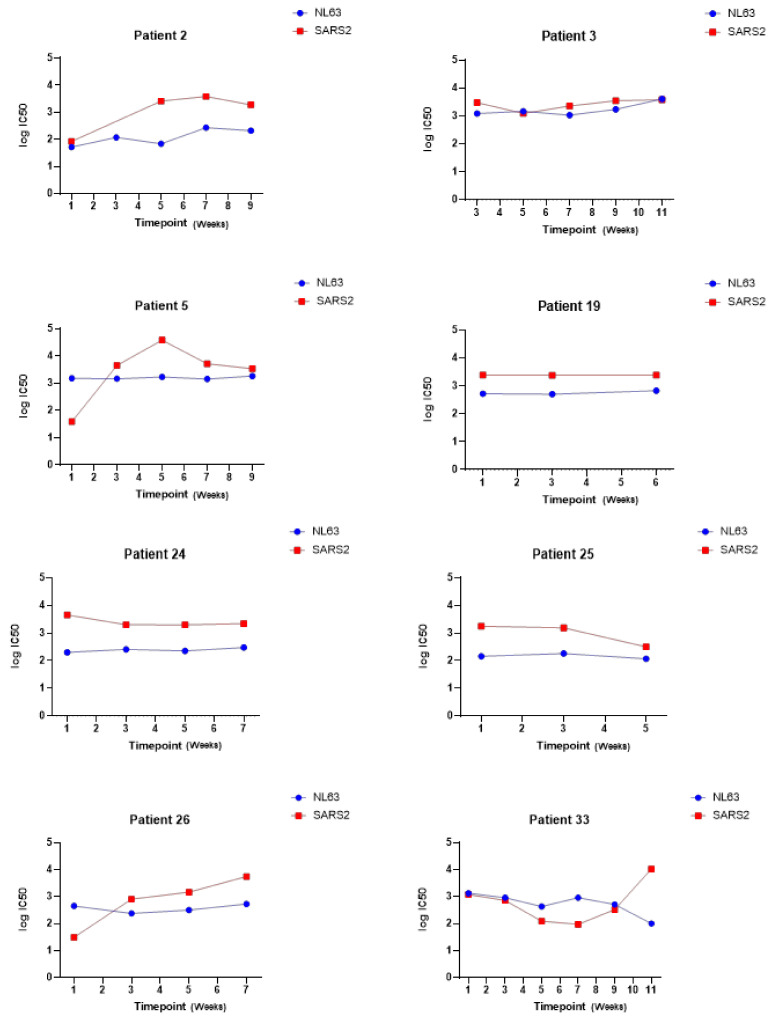
Example data showing log IC50 values for neutralization of HCoV-NL63 and SARS-CoV-2 at sequential time points in a cohort of patients hospitalized with COVID-19; samples were collected through the HICC consortium (https://www.hicc-consortium.com/, accessed on 10 May 2021). Serum samples were collected weekly with timepoint 1 representing a serum sample taken at admission.

**Table 1 viruses-13-01579-t001:** Receptor usage varies between human coronaviruses.

HCoV	Cellular Receptor
HCoV-229E	Aminopeptidase N
HCoV-NL63	ACE2
SARS-CoV-1	ACE2
SARS-CoV-2	ACE2
MERS-CoV	DPP4
HCoV-OC43	9-O-Ac-sialic acid
HCoV-HKU1	9-O-Ac-sialic acid

## Data Availability

Data and plasmids available from corresponding authors via email request.

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
