# Peer review of "Coronavirus Pseudotypes for All Circulating Human Coronaviruses for Quantification of Cross-Neutralizing Antibody Responses"

_viruses, 2021, doi:10.3390/v13081579_

Round 1

Reviewer 1 Report

This is an interesting work describing a panel of lentiviral pseudotypes bearing the spike proteins for each of the human coronavirues (both those associated to 'common cold' and the three causing severe disease) . This methodology is particularly important  for quantification of cross-neutralising antibody responses and in my opinion the work could be accepted for publication after minor revisions.

1-This is a generally well written manuscript. Nonetheless, I suggest the authors to check carefully their manuscript because for minor mistakes as there could be points needing to be clarified (line 458, p. 12: 'at higher levels that HEK293T/17 cells,' may be than in place of that? ; etc).

2-empty space should be reported between numbers and uM, uL (u=micro), ng, ug etc especially in materials and methods, but also in other parts of the manuscript

3- more updated numbers on COVID-19 should be provided as sadly the deaths are have reached 3.3 millions.

4-more details on the vaccines already approved or under development should be added to the revised manuscript. Some information on what we know presently on the durability of the response after anti-SARS-CoV-2 vaccination should be provided discussing works like DOI 10.1056/NEJMc2032195 or others of your choice

5-the importance of repurposed drugs in the fight against COVID-19 and other highly pathogenic coronaviruses should be mentioned citing at the least the work DOI:  10.3390/molecules26040986

6-Figure 1: after 'D' one would expect another arrow with something written next: may be ' non-human coronaviruses'? or something like that.

7-all abbreviations should be explained at their first usage in the manuscript.

8-I would use SARS-COV-1 and SARS-CoV-2 in place of SARS1 and SARS2 in figure 7 and 8. Otherwise explain the meaning of these abbreviations in the legend of fig. 7

9-line 191 of section 2.2: 37°C : C should not be superscript

10-Fig. 10: what about the neutralization data for the other 4 HCoVs using the SARS-CoV-2 seropositive serum? Is there something to show in Supporting Information or in the revised Figure 10? 

11-Is it feasible to perform similar studies (with respect to those depicted in Fig 10) using 'common cold' HCoV seropositive sera to see for example the effect had against SARS-CoV-2 and MERS-CoV? Please discuss and/or better explain this point, also as a perspective in continuing the current research in view of a future article. Is there anything similar in the literature, i.e. on the possible protective role against SARS-CoV-2 of antibodies in serum of common cold HCoV seropositive individuals?

12-The section Funding and some of the other that follow it should be revised.

Author Response

Please see the attached response from the authors.

Reviewer 2 Report

The authors developed a microplate assay to test the capacity of antibodies and sera using pseudotyped viruses reflecting all seven human CoVs. These days these kind of tools are likely to become elementary to identify crossprotection of existing antibodies due to suffered illness or if vaccination strategies lead to antibodies with crossprotecting capacities. Thus, the new assay would be of great help to answer currently important questions. While the idea and the concept are elegant and of major interest to the field, there are some major issues regarding its presentation and results.

  • The authors used only two sera to investigate crossprotection using their assay. The manuscript would greatly benefit from additional donors to show the capacity of the assay. The authors should show results using sera of several donors with known infection of the endemic CoVs, of SARS-CoV1, SARS-CoV-2 and MERS-CoV as well as from naïve donors to present the full capacity of the assay. Why developing OC43 PVs and then not using them? The authors should put really much more emphasis on this, to show the power and the potency of the assay.
  • Data in Figure 9 are of poor quality, please provide pictures of high quality, where cells can be identified and background is reduced. Please provide quantitative data.
  • Why presenting GFP reporter PVs if the final assay is made with luciferase-reporter PVs? Do the results look the same in both cases? Please provide comparative data using both type of PVs or just provide data on one type.

Minors:

  • y axis labeling can be improved, a linear scale together with box and whiskers blots showing all data points will greatly support the credibility and visibility of the data
  • are there any significant differences? please provide Information on the statistics and data/graphic preparation
  • It would make sense to condense the development of the assay in one figure and one paragraph
  • The use of RRID identifiers is strongly recommended.
  • Check if all annotated figures are in bold letters.

Author Response

(The authors gave the same response as above.)

Reviewer 3 Report

The authors describe the development of a panel of lentiviral pseudotypes carrying the Spike protein of all known human coronaviruses. The manuscript is very well written and easy to follow. However, this study requires major improvements before it can be considered for publication.

Major points:

  1. The authors nicely describe the development of a panel of Spike-bearing lentiviral pseudotypes of all known human coronaviruses and thus state that the panel can be used for the analysis of cross-neutralizing antibody responses in the title, results, and discussion. However, the authors only describe the development of the panel. Indeed, they used only TWO human sera for their demonstration. First, these sera are not sufficiently described in the method section and are not fully characterized. In addition, the use of only two samples is not statistically significant. To ensure that the pseudotype panel can be used for the analysis of cross-neutralization antibody responses, the authors should use a sufficient number of samples that are fully characterized with an alternative reference assay (i.e., by comparison with seroneutralization using the corresponding wild-type viruses) and especially to have positive and negative sera for each coronavirus.
  2. The authors also claim that their panel of pseudotypes can be used in high-throughput assays, but nothing like this has been demonstrated. Did the authors perform the experiments with robots that could allow automation of the process? What was the throughput? The authors also showed the need to improve their system by transfecting receptors or accessory cellular proteins into target cells. This method was already known and others had already shown it to improve viral replication. Target cell transfection is obviously difficult to implement in a high-throughput setting and it is even more difficult to standardize the experiment. The authors should consider these factors and also describe all the advantages and disadvantages of their panel.

Minor points:

Line 42-45: Authors should refresh the statistics.

Line 94: Fig 1a refer to nothing existing in the manuscript

Line 190: 37°C should be 37°C

Line 328-334: The authors should describe here all the cells used for permissive testing and better describe the results or present all the data as supplementary data for example. When only one cell line is presented Fig 5a, obviously it is "the most permissive cell line" and this is therefore confusing for readers.

Line 419: "In Figure 6" should be corrected

Author Response

(The authors gave the same response as above.)

Reviewer 4 Report

The authors optimized conditions for the production of high-titer lentivectors pseudotyped with human coronavirus Spike in HEK293T/17 cells.

They generated a panel of 7 different constructs pseudotyped with: HCoV-229E, HCoV-NL63, SARS-CoV-1, SARS-CoV-2, MERS-CoV, HCoV-OC43 and HCoV-HKU1. Since these different coronaviruses use different receptors to infect cells the authors detected the functional titers transducing different cell line. Pseudotyped lentivirus preparations were used  to measure neutralizing and cross-neutralizing activity of two !?!? sera from SARS-CoV-2 infected patients.

Pseudoviruses are an essential tool to study viruses. Several groups have proposed similar approaches to address SARS-CoV2 Spike and other coronaviruses, therefore the novelty of the work is very minor. Furthermore, if the intention is to use them to quantify cross-neutralizing antibody responses, as suggested in the title, they will need to provide quantitative data on a robust panel of sera. Finally, the manuscript is poorly written, with several details missing that makes it difficult to interpret the data, below some examples. To conclude, as a purely methodological paper is of little interest and lacks precision. As a scientific paper it does not explore a scientific question.

Comments:

1) The authors use the 7 panel of coronaviruses to test SARS-CoV-2 antibody neutralization and cross-neutralization activities.

The principle is that neutralizing antibodies bind Spike blocking the interaction with the receptor or preventing/inhibiting the membrane fusion.

The neutralization activity depends also by the antibody titer. In this system the author do not evaluate the sensitivity of their platform, for this, they should compare the same panel of sera in neutralization and cross-neutralization using true coronaviruses (Plaque Reduction Assay).

The number of seropositive and seronegative sera tested should be reported, if more than the 2 shown. if these are the only one this is insufficient.

2) The authors produced lentiviruses pseudotyped with SARS-CoV-2 Spike-Envelope and Membrane proteins. They should use this preparation, in comparison with lentivirus pseudotyped with only SARS-CoV2 Spike glycoprotein to quantify neutralization. Also, most groups use a delta19 version of Spike to increase transduction, how does this compare with their method?

3) In Figure 5 panel c - a legend should be inserted.

4) From line 316 to 318. No matches between the text and the figure 5b and 5C

5) In the Figure 8a the authors reported that optimal amount of plasmid coding for ACE2 is 1 mg  (line 377) while from plot analysis it looks 500 ng.

6) The authors produced GFP-bearing lentiviruses. Transduction titers should be detected.

7) In figure 9 (line 399), the authors refer to transfected cells but they show a transduction experiment.

8) In line 203 should be written mm and not mM.

9) Figure 10 legend, 190 and 193 appear inverted in the description.

10) SARS-CoV-2 variants can be engineered in this context, any data in this direction?

Author Response

(The authors gave the same response as above.)

Round 2

Reviewer 2 Report

The authors provided a revised manuscript and included new data to illustrate the potential of their assay. The inclusion of several sera to demonstrate the neutralization capacity improves the quality of the manuscript. However, there are some residual issues.

Major:

- The GFP reporter PVs were not used for functional assays and the quality of the images regarding background is still poor. On the other side, all functional assays were conducted using the luc reporter. Please provide pictures of high quality, where cells can be identified and background is reduced and functional data with the GFP-PVs, or the authors may consider to remove the data, as they are not required for the residual manuscript.

- the authors could discuss, if their assay has potential to provide a correlate of immunity compared to serological antibody tests

- the authors should provide a restructured figure 9 in colour and fitting on one page.

- please provide information in figure legends about the cohort size, f.i. in fig 9c, representative results of 8 from ? sample are shown...or in fig 7 results are shown as mean +/- SD? Of three independent experiments run in triplicates? ...

Minor

- please provide a time unit in the x axis fig 9c

- please check for spaces

Author Response

The authors provided a revised manuscript and included new data to illustrate the potential of their assay. The inclusion of several sera to demonstrate the neutralization capacity improves the quality of the manuscript. However, there are some residual issues.

The authors would like to thank the reviewer for their comments and thorough review. The authors believe we have addressed the required corrections, below.

Major:

- The GFP reporter PVs were not used for functional assays and the quality of the images regarding background is still poor. On the other side, all functional assays were conducted using the luc reporter. Please provide pictures of high quality, where cells can be identified and background is reduced and functional data with the GFP-PVs, or the authors may consider to remove the data, as they are not required for the residual manuscript.

  • Point 1 – We have removed this section from the manuscript

- the authors could discuss, if their assay has potential to provide a correlate of immunity compared to serological antibody tests

  • Point 2 – Line 474, p. 12 – we have discussed the potential use of the assay in providing a correlate of immunity.

- the authors should provide a restructured figure 9 in colour and fitting on one page.

  • Point 3 – Figure 9 and 10, pages 10 and 11 – we have restructured the original figure to fit one a single page and adding colour for clarity.

- please provide information in figure legends about the cohort size, f.i. in fig 9c, representative results of 8 from ? sample are shown...or in fig 7 results are shown as mean +/- SD? Of three independent experiments run in triplicates?

  • Point 4 – These are example data from an ongoing study as part of the HICC consortium (https://www.hicc-consortium.com/) – the patient cohort currently numbers over 200 but is still recruiting.

Minor

- please provide a time unit in the x axis fig 9c

  • Point 5 - Figure 10, page 11 – we have added a time unit in the x-axis

- please check for spaces

Thank you.

Reviewer 3 Report

The authors have improved the quality of the manuscript which is now acceptable for publication

Author Response

The authors have improved the quality of the manuscript which is now acceptable for publication

The authors would like to thank the reviewer for their comments and time reviewing the article.

Reviewer 4 Report

The revised version is improved mostly by the increase of the panel of sera tested for neutralization. surprisingly the sera that were all from COVID patients were not tested for SARS-CoV-2 neutralization. What are the authors concluding from this experiment? that the serum of SARS-CoV-2 infected individuals is not able to neutralize the other members of the family? I believe this part is incomplete and needs to show neutralization of SARS-CoV-2 PV at least to demonstrate the validity of these PV. For a better work mice could be immunized with the various PV thus providing clean sera to check cross-reactivity, but I understand it will be a lot of work.

Author Response

The revised version is improved mostly by the increase of the panel of sera tested for neutralization. surprisingly the sera that were all from COVID patients were not tested for SARS-CoV-2 neutralization. What are the authors concluding from this experiment? that the serum of SARS-CoV-2 infected individuals is not able to neutralize the other members of the family? I believe this part is incomplete and needs to show neutralization of SARS-CoV-2 PV at least to demonstrate the validity of these PV. For a better work mice could be immunized with the various PV thus providing clean sera to check cross-reactivity, but I understand it will be a lot of work.
The authors would like to thank the reviewer for their comments and thorough review. 
Point 1 - We have updated the manuscript with corresponding neutralisation data for the same serum samples against SARS-CoV-2, demonstrating neutralisation of our SARS-CoV-2 PVs by these samples. 
Point 2 - Whilst we agree that ‘clean’ antisera against the individual coronaviruses would be desirable, immunisation of mice with our PV panel is currently outside our capabilities; nevertheless, since this limitation exists for wild-type virus assays and serological assays such as ELISA, we believe this panel of PVs is still a comparably useful tool in the investigation of cross-reactivity.